# STAT3: A Promising Therapeutic Target in Multiple Myeloma

**DOI:** 10.3390/cancers11050731

**Published:** 2019-05-26

**Authors:** Phyllis S. Y. Chong, Wee-Joo Chng, Sanjay de Mel

**Affiliations:** 1Cancer Science Institute of Singapore, National University of Singapore, Singapore 117599, Singapore; csicsyp@nus.edu.sg (P.S.Y.C); wee_joo_chng@nuhs.edu.sg (W.-J.C.); 2Department of Haematology-Oncology, National University Cancer Institute of Singapore, National University Health System, Singapore 119074, Singapore

**Keywords:** multiple myeloma, STAT3, targeted therapy

## Abstract

Multiple myeloma (MM) is an incurable plasma cell malignancy for which novel treatment options are required. Signal Transducer and Activator of Transcription 3 (STAT3) overexpression in MM appears to be mediated by a variety of factors including interleukin-6 signaling and downregulation of Src homology phosphatase-1 (SHP-1). STAT3 overexpression in MM is associated with an adverse prognosis and may play a role in microenvironment-dependent treatment resistance. In addition to its pro-proliferative role, STAT3 upregulates anti-apoptotic proteins and leads to microRNA dysregulation in MM. Phosphatase of regenerating liver 3 (PRL-3) is an oncogenic phosphatase which is upregulated by STAT3. PRL-3 itself promotes STAT-3 phosphorylation resulting in a positive feedback loop. PRL-3 is overexpressed in a subset of MM patients and may cooperate with STAT3 to promote survival of MM cells. Indirectly targeting STAT3 via JAK (janus associated kinase) inhibition has shown promise in early clinical trials. Specific inhibitors of STAT3 showed in vitro efficacy but have failed in clinical trials while several STAT3 inhibitors derived from herbs have been shown to induce apoptosis of MM cells in vitro. Optimising the pharmacokinetic profiles of novel STAT3 inhibitors and identifying how best to combine these agents with existing anti-myeloma therapy are key questions to be addressed in future clinical trials.

## 1. Introduction

### 1.1. Multiple Myeloma

Multiple myeloma (MM) is the second most common haematological malignancy in adults worldwide [1]. MM is characterised by the accumulation of malignant plasma cells in the bone marrow and is usually associated with a monoclonal protein. Anaemia, lytic bone lesions, renal impairment and immune paresis are common clinical findings in MM [2]. A variety of genetic subtypes have been described which correlate with clinical behaviour as well as treatment response and prognosis [3]. Translocations (4;14), (14;16) as well as, 17p13 deletion and 1q21 amplification, are associated with an adverse outcome and the optimal management of this group of patients remains to be established [3,4]. Treatment for MM has evolved significantly over the last two decades and patients now have access to effective therapeutic agents with acceptable toxicity profiles [5]. In particular, the advent of proteasome inhibitors (PI), immuno-modulators (IMID), and monoclonal antibodies have resulted in significant improvements in survival for patients with MM [6]. Unfortunately MM remains incurable, with relapse occurring in virtually all patients despite novel agent-based therapy [1]. There is hence an urgent need to identify novel treatment options with the endeavour to find a cure for MM.

### 1.2. Overview of the STAT Transcription Factor Family in Oncogenesis

Signal transducers and activators of transcription (STAT) have been recognised for over two decades as a family of transcription factors critical for a variety of cellular functions including cell survival, proliferation, angiogenesis, invasion and metastasis [7]. Seven members of the STAT protein family (STAT-1,2,3,4,5a, 5b and 6) are currently known, they reside in the cytoplasm and are activated by extracellular factors to elicit an intracellular response through regulating gene expression [8]. Growth factor receptors or oncogenic kinases such as Janus Associated Kinases (JAK), epidermal growth factor receptor (EGFR), Src family members and Bcr-Abl have been reported to activate STAT proteins [9]. In MM, interleukin 6 (IL-6) signalling in the bone marrow leads to activation of the JAK-STAT pathway resulting in phosphorylation of STAT proteins [9]. Upon phosphorylation, the STAT dimers translocate to the nucleus and regulate transcription [10].

The STAT transcription factors play a physiologic role as mediators of cell proliferation and growth [8]. In particular, they play a pivotal role in normal haematopoiesis and immune development [11]. Deregulation of STAT signalling also plays an important part in the pathogenesis of a variety of malignancies such as breast and lung cancer as well as melanoma [12]. Among the STAT proteins, STAT3 and STAT5 are the most strongly implicated in the pathogenesis of cancer [13]. Not only are they transducers of many important oncogenic signalling pathways, through their role as transcription factors they also modulate the expression of a variety of genes involved in neoplasia [12]. STAT3 regulates genes by binding to their promoter region through recognizing the consensus site TTN_5_AA (i.e., TTNNNNNAA) [14]. Potent proto-oncogenes like c-myc, cyclin D1 and anti-apoptotic genes Mcl-1, Bcl-xL and Bcl-2 were identified as downstream targets of STAT3 [14]. There is also emerging evidence that STAT signalling impacts the tumour microenvironment and may play a role in tumours escaping immune surveillance [12,15,16]. CD8^+^ T effector cells are an important component of both the innate and adaptive anti-tumour response [15]. Activation of STAT3 has been shown to negatively regulate the cytotoxic T-cell mediated immune response [15]. In a mouse model of melanoma, it was further demonstrated that silencing STAT3 in the myeloid and B cells of the tumour microenvironment could circumvent the immunosuppressive effects of STAT3 and achieve clearance of the malignant cells [15].

### 1.3. STAT3 Activation Is an Unfavourable Prognostic Factor in MM

STAT3 expression has been shown to have an adverse prognostic impact in MM. In a cohort of newly diagnosed MM patients, tyrosine phosphorylation of STAT3, which is a marker for STAT3 activation, was detected by immunohistochemical staining (IHC) in 10 out of 94 patients (10.6%) [17]. Patients with STAT3 activation were found to have significantly shorter progression-free and overall survival. In another study, activation of STAT3 could be detected in almost half of 48 MM patients, survival data for this cohort were however not available [18]. Preclinical data also suggest that STAT3 activation may promote drug resistance [17,19].

The role of STAT3 in malignancy, along with its prognostic impact has generated significant interest in STAT3 as a therapeutic target in MM. In this review we will discuss the key aspects of STAT signalling in the pathogenesis of MM and highlight its potential as a novel therapeutic target for this incurable malignancy.

## 2. Mechanisms of STAT3 Activation in MM

The importance of STAT3 in MM was highlighted by the finding of constitutively activated STAT3 in CD138+ bone marrow cells isolated from the majority of MM patient samples, but not in healthy controls [20,21]. The finding of uniform STAT3 activation despite the known heterogeneity of MM supports its potential as a therapeutic target. Furthermore, the majority of MM cell lines are responsive to IL-6 treatment and consequently upregulate STAT3 phosphorylation [22] As mutations in the *STAT3* gene were not observed in MM, the hyperactivation of STAT3 can be attributed to various genetic and epigenetic mechanisms. The mechanisms leading to STAT3 activation in MM are summarised in Figure 1 and provide alternative examples of strategies for indirect inhibition of STAT3.

The canonical IL-6 signalling is tightly regulated by the repressors Src homology containing protein 1 (*SHP-1)* and *SHP-2* tyrosine phosphatases, together with suppressor of cytokine signalling 1 (*SOCS1)*. The disruption of this negative feedback loop results in a hyper-stimulated JAK/STAT pathway. Indeed, the expression of these genes were found to be silenced in malignant plasma cells, and this correlated with increased JAK/STAT3 activity [23,24]. In a pivotal study, Chim et al. demonstrated that hypermethylation of *SHP-1* is frequently observed in MM (79.4%, n = 34) [25]. Using the human MM cell line U266 which displayed complete methylation of *SHP-1*, they further demonstrated that 5-azacytidine could reverse the methylation and restore the expression of *SHP-1*. This corresponded with a down-regulation of STAT3 phosphorylation, demonstrating that epigenetic downregulation of *SHP-1* results in activation of STAT3. Likewise, *SOCS1* was also found to be silenced by hypermethylation in 62.9% (n = 35) of MM patient samples [24]. In contrast, hypermethylation of *SOCS-1* was absent in normal plasma cells. Considering these findings, demethylating agents may warrant evaluation as a therapeutic modality in STAT3 overexpressing MM.

Nuclear receptors are a distinct class of transcription factors that respond to hormones and ligands, and have been reported as negative regulators of IL6-mediated STAT3 activity [26,27]. Binding of estrogen to estrogen receptors (ER) induced the transcription of PIAS3 (protein inhibitor of activated STAT3), which interacts with STAT3 at its DNA-binding domain to block STAT3 mediated activation of target genes [26], without affecting the activation of JAK2 or phosphorylation of STAT3. Treating MM cells with estrogen or estrogenic ligands abolished IL-6 induced MM cell proliferation. Another nuclear receptor, peroxisome proliferator-activated receptor (PPARγ) was also implicated as a regulator of IL6-STAT3 signalling. PPARγ ligands, but not PPARα, suppressed IL-6-mediated MM cell growth in PPARγ-positive MM cell lines [28]. Additionally, this inhibition could be observed in MM cell lines which are responsive but do not depend on IL-6 for survival. More importantly, PPARγ ligands inhibited the growth of patient derived MM cells. Mechanistically, the inhibition of cell growth was demonstrated to be associated with the downregulation of the expression of *c-myc*, *mcl-1*, *bcl-xl*, *bcl-2*, and upregulation of pro-apoptotic genes *bax* and *bak*. Ligand-activated PPARγ reduced binding of STAT3 to the SIE(c-sis-inducible element) as demonstrated by electrophoretic mobility shift assay (EMSA), and luciferase reporter assay further revealed the abrogation of the DNA binding ability of STAT3. Specifically, in the presence of PPARγ ligands, the occupancy of STAT3 on *c-myc* and *mcl-1* promoters were abolished. The involvement of at least two different classes of nuclear receptors in supressing STAT3 function suggest potential therapeutic implications.

Constitutive activation of STAT3 could be the result of dysregulated upstream signalling from JAK2 [29]. A gain-of-function mutation at position 617 from valine to phenylalanine in the JH2 domain of JAK2 (V617F) leads to hypersensitivity towards cytokine stimulation and constitutive activity of the kinase. This mutation is well recognised in myeloproliferative neoplasms [30,31]. The *JAK2*^V617F^ allele, however, was not found in lymphoid malignancies or multiple myeloma [31,32,33]. Despite the absence of mutations, overexpression of *JAK2* was reported in 57% of MM patients [34]. *JAK2* overexpression was also demonstrated in MM cells with miRNA-375 promoter hypermethylation [35]. Aberrant repression of miRNA-375 was found in plasma cells from monoclonal gammopathy of undetermined significance (MGUS), newly diagnosed and relapsed MM patients. Restoration of miRNA-375 expression using hypomethylating agents or histone deacetylase inhibitors could modify *JAK2* levels [35].

Post-translational modification through reversible methylation of STAT3 by histone-modifying enzymes has been reported in other cancers [35,36,37]. Dimethylation at lysine 140 is catalysed by SET9 (SET domain containing lysine methyltransferase 9) and lysine 49 by EZH2 (enhancer of zeste homolog 2), while trimethylation of lysine 180 is mediated by phospho-EZH2 [35,36,37]. These methylation events were demonstrated to modulate the IL-6 response and transcription of target genes [36]. Apart from phosphorylation and methylation, other post-translational modifications like acetylation, SUMOylation, S-nitrosylation and ubiquitination are reported to regulate the functions of STAT3 in multiple cancer types [38]. Acetylation of STAT3 has been reported to promote its transcriptional activity, especially on lysine 685 [39,40]. STAT3 also binds to protein inhibitor of activated STAT(PIAS) proteins (SUMO E3 ligases), and the interaction can be promoted by cytokines such as IL-6, implicating a role of SUMOylation in cytokine signaling [41]. In MM cell lines, S-nitrosylation of STAT3 negatively regulates its activities resulting in cell cycle arrest and could rescue MM cells from melphalan induced cytotoxicity [42]. Activation of the E3 ubiquitin ligase c-Cbl leads to the proteasomal degradation of SOCS3 (suppressor of cytokine signalling3), promoting STAT3 activity [43]. The relevance of these modifications in MM should be evaluated in future studies.

While the canonical functions of STAT3 as a transcriptional regulator are well-studied, findings that STAT3 can localize to the mitochondria (mitoStat3) in multiple cell types, including MM, revealed its regulatory activities in cellular metabolism [44]. The interacting partners of STAT3 in the mitochondria are implicated in mitochondrial translation and electron transport chain (ETC) pathways. It was also reported that mitoSTAT3 is essential for Ras-mediated cellular transformation, and blockade of mitoSTAT3 leads to anti-tumor activity. Moreover, these processes were dependent on the phosphorylation of STAT3 on serine 727, which can be induced by IL-6. Given the importance of the IL6-STAT3 signaling pathway in the survival of myeloma cells, it will be interesting to study mitoSTAT3 in the context of myeloma. Additionally, the role of STAT3 in mediating drug resistance was also suggested to be attributed to mitoSTAT3 through resistance to oxidative stress. Collectively, these findings suggest that targeting mitoSTAT3 should also be considered in addition to targeting the transcriptional activities of STAT3. The finding that miR551b and high glutamate levels were associated with activation of STAT3 in invasive ovarian cancer further support the hypothesis that STAT3 has a role in cellular metabolism [45].

## 3. Downstream Targets of STAT3 Implicated in Myeloma Cell Survival

STAT3 is a central regulator of IL-6 signalling, activating both immediate/early IL-6 responsive genes, as well as genes involved in the prolonged response towards IL-6 [22]. STAT3 elicits a pro-survival status in MM cells largely through suppression of apoptosis. Hence, members of the Bcl-2 anti-apoptotic family have been implicated as important mediators of IL-6-dependent survival [21,46,47,48]. STAT3 binding motifs were found in the promoter region of Bcl-xL [21] and Mcl-1 [49], indicating that STAT3 could regulate their expression as a transcription factor. The U266 human MM cell line, which displays high levels of activated STAT-3 and Bcl-xL, is intrinsically resistant to Fas-mediated apoptosis [21]. Further studies have suggested that Mcl-1 may play a more essential role than Bcl-xL or Bcl-2 in MM cells dependent on IL-6 [46,47].

While upregulation of anti-apoptotic genes is important for myeloma cell survival, it does not fully account for the survival benefits conferred by IL-6 through STAT3. In the microarray dataset reported by Brocke-Heidrich et al. it is interesting to note that *PTP4A3*, a gene encoding for the phosphatase of regenerating liver 3 (PRL-3), was identified as an IL-6-responsive gene [22]. PRL-3 is a potent oncogenic phosphatase with an important role in metastasis and disease progression [50]. The overexpression of PRL-3, driven by numerous regulatory mechanisms, could be observed across different solid tumours and haematological malignancies [51]. The oncogenic properties of PRL-3 are attributed to its ability to dephosphorylate a large number of signalling proteins, through its conserved C(X_5_)R (e.g., CXXXXXR) catalytic domain [52]. In addition to IL-6, IL-15 and IL-21 were also reported to induce PRL-3 expression, suggesting the importance of PRL-3 as a converging signal of different growth-promoting cytokines in MM [53]. Furthermore, ectopic expression of PRL-3 in INA-6 could promote cell survival and rendered the cells IL-6-independent [54]. In our previous report, we identified a conserved and functional STAT3 binding motif in the promoter of PRL-3, indicating that STAT3 is a direct upstream regulator of PRL-3 [55]. To further support our notion, PRL-3 expression significantly correlates with a STAT3 activation signature in acute myeloid leukaemia (AML) datasets [55]. On the other hand, we and others have observed that PRL-3 promotes STAT3 phosphorylation [54,55], suggesting that a positive feedback loop exists between PRL-3 and STAT3 to drive a potent pro-survival signal. While STAT3 directly regulates PRL-3 at the transcriptional level, how PRL-3 promotes the phosphorylation of STAT3 remains to be determined. Importantly, elevated expression of *PTP4A3* was observed in a subset of patient derived MM cells. [56].

STAT3 was reported to play a role in the regulation of microRNA-21 (miR-21) and long non-coding RNAs (lncRNAs). miRNAs are a class of short, non-coding RNA that bind to target mRNAs and mediate their degradation at the post-transcriptional level. There is growing evidence implicating miRNA in the pathogenesis, progression and drug resistance of MM. [57,58,59,60,61]. The enhancer region of *miR-21* contained two consensus STAT3 binding sites and is evolutionally conserved across all vertebrates. To determine whether the STAT3-binding site of *miR-21* is functional, IL-6 treated cells were subjected to chromatin immunoprecipitation and reporter assay which demonstrated STAT3 recruitment to the upstream region of *miR-21* to regulate its expression [57]. As miR-21 is an anti-apoptotic factor in cancer cells [61], STAT3-mediated expression of miR-21 could partly explain the IL-6-induced survival of myeloma cells. Additionally, increased expression of miR-21 could confer cytokine independence from IL-6.

Another novel class of non-coding RNAs, the lncRNAs, consist of >200 nucleotides with a functional role in gene regulation [62]. The emerging role of lncRNAs in MM was discovered when lncRNA metastasis-associated lung adenocarcinoma transcript 1 (*MALAT1*) was found to be overexpressed in newly diagnosed MM, but not in post-treatment patients and healthy controls [63]. Another study was carried out to profile differentially-regulated lncRNAs in newly diagnosed MM, which led to the identification of ST3GAL6 (ST3 beta-galactoside alpha-2,3-sialyltransferase 6)-AS1, LAMA5-AS1 and RP11-175D17.3 [64]. Interestingly, tiling arrays revealed that IL-6 also induced the transcription of a number of lncRNAs through STAT3, and the authors termed these transcripts STAT3-induced ncRNAs (non-coding RNAs), in short STAiRs [64]. Specifically, STAiRs 1, 2, 6, 15 and 18 were described to be important lncRNAs mediated by STAT3. While STAiR18 is a potential epigenetic regulator, STAiR2 was found to regulate its host gene, the tumour suppressor named Deleted in Colorectal Cancer (*DCC)*. The tissue-specific expression of these STAiRs were also analysed and STAiRs 1 and 2 could serve as myeloma-specific markers. Further studies are required to elucidate how the alterations in lncRNAs promote the progression of MM.

14-3-3 proteins bind to an array of serine/threonine phosphorylated proteins, playing the role of adaptor or scaffold proteins in signal transduction pathways [65]. Serine 727-phosphorylated STAT3 was found to physically associate with 14-3-3ζ in a proteomics screen for novel cancer-associated 14-3-3ζ-binding partners [66,67]. While phosphorylation of STAT3 on tyrosine 705 is well-elucidated and functionally important, 14-3-3ζ modulation of STAT3 activity through its serine 727 phosphosite is as important since concurrent phosphorylation of both tyrosine 705 and serine 727 is required for the full activation of STAT3 [68]. It was further shown that 14-3-3ζ protects STAT3 from protein phosphatase 2A (PP2A)-mediated dephosphorylation, prolonging the signalling response, and is required for the nuclear translocation and transactivation activities of STAT3. While therapeutic targeting of 14-3-3ζ is premature due to its involvement in cellular homeostasis, abrogation of the interaction between STAT3 and 14-3-3ζ can be considered as a potential therapeutic strategy.

## 4. Strategies for STAT3 Inhibition in MM

Modalities for targeting STAT3 in cancer may be classified into direct and indirect approaches [10]. Agents directly targeting STAT3 may either inhibit dimerization of the STAT proteins, inhibit STAT3 mRNA or interfere with STAT3 DNA binding [10]. Indirect approaches include preventing ligands binding to growth factor receptors, inhibiting upstream tyrosine kinases such as JAK or activating negative regulators of STAT3 [10]. An overview of the STAT3 inhibitors evaluated against MM is presented in Table 1.

### 4.1. Indirect STAT3 Inhibition

The JAK inhibitor ruxolitnib is approved for the treatment of patients with myelofibrosis [69]. Given the overexpression of JAK 1 and 2 in MM, there has been increasing interest in evaluating JAK inhibition as a therapeutic target. D’Oliveira and colleagues showed that ruxolitinib as a single agent had no effect on MM cell lines but induced apoptosis when combined with bortezomib [33]. Co-culture with stromal cells rescued the MM cells from apoptosis but this effect was ameliorated by the addition of lenalidomide [33]. These findings suggest that the microenvironment protects MM cells from JAK inhibition. This may be explained by the stromal cells providing an alternative means of STAT activation. The cytotoxic effect of lenalidomide in this setting maybe occurring through its immunomodulatory function and further studies are required to better understand the mechanisms involved.

Chen et al. demonstrated that ruxolitinib had single agent activity against MM cell lines as well as primary MM cells. There was also synergy with lenalidomide and dexamethasone. This combination was also effective in patient derived xenografts as well as a patient with concurrent polycythaemia vera and MM [70]. A phase 1 study has recently reported encouraging data that ruxolitinib may restore sensitivity to lenalidomide in MM patients who were previously lenalidomide refractory [71]. Larger trials are required to definitively assess the role of ruxolitinib in MM. An important question to be answered by these studies is whether sensitivity to ruxolitinib is limited to JAK overexpressing MM.

Through a drug repurposing screen, Lam et al. identified tofacitinib, a JAK inhibitor approved for rheumatoid arthritis to have activity against MM cells. The activity of tofacitinib appeared greater in an IL-6 dependent MM cell line and was particularly effective at inhibiting stromal cell derived growth signals [72]. Interestingly ruxolitinib was shown to have very limited activity against MM cell lines in this study. Intriguingly, tofacitinib synergised with venetoclax to induce apoptosis of MM cells in co culture with stromal cells but not in monoculture [72]. These data highlight the importance of stromal growth signalling in MM and call for further studies to evaluate the mechanisms behind this unexpected synergy.

The pan JAK inhibitor INCB20 has also shown promising in vitro efficacy against MM cell lines which was sustained in the presence of bone marrow stromal cells [73]. IL-6 is known to impair dexamethasone induced apoptosis of MM cells. Treatment with INCB20 abrogated the protective effect of IL-6 and sensitised MM cells to dexamethasone [73]. Importantly, STAT3 phosphorylation was markedly reduced in INCB20 treated cells. These data support the hypothesis that JAK signalling as well as IL-6 are important for MM cell survival and that STAT3 can be effectively targeted by inhibition of JAK. Given the broader spectrum of JAK inhibition by INCB20, clinical trials evaluating this compound are important to compare the toxicities and off target effects with more specific JAK inhibitors such as ruxolitinib.

A JAK inhibitor with greater specificity for JAK1 and 2 (INCB16562) was also shown to induce apoptosis in MM cell lines [74]. As seen with the INCB20, IL-6 mediated survival of MM cells was diminished and sensitivity to bortezomib and melphalan were increased by treatment with INCB16562 [74]. These findings were also confirmed in patient derived xenograft (PDX) models. This data suggests that JAK1 and JAK2 maybe the key JAK family kinases involved in MM survival and that pan JAK inhibition may not be required to achieve cytotoxicity. Further data supporting more selective JAK inhibitors were presented by Scuto, et al. who demonstrated that the selective JAK2 inhibitor AZD1480 was able to inhibit STAT3 phosphorylation and induce apoptosis in MM cell lines as well as in PDX [75]. Another novel JAK inhibitor NS-018 was not only effective at inhibiting IL-6 mediated survival of MM cells but also abrogated osteoclast activation in a murine MM model, likely through its activity against the Src kinase. These findings suggest that JAK inhibition has significant potential as a therapeutic strategy in MM. Identification of the genetic subtypes of MM most susceptible to JAK inhibition as well as studying how JAK inhibition may combine with current therapeutic modalities in MM are challenges to be addressed by future studies. Ramakrishnan et al. reported synergism between JAK2 inhibition and the pro-apoptotic caspase activator LCL-161 against MM cell lines [76]. Interestingly LCL-161 by itself did not induce apoptosis in all MM cell lines, suggesting there is an interplay between JAK signaling and apoptotic pathways in MM.

The antihelminth drug niclosamide was identified through a drug redeployment screen to have anti myeloma activity [77]. Niclosamide induced apoptosis in MM cell lines through multiple mechanisms including inhibiting IL-6 mediated STAT3 phosphorylation. Although synergy with known anti myeloma drugs was not assessed in this study, niclosamide showed a similar potency in vitro to some anti myeloma agents. The established safety profile of this drug in humans makes it an attractive option for evaluation in clinical trials.

### 4.2. Direct STAT3 Inhibition

Sinomenine is a derivative of a medicinal plant sinemeniumacatum which has been used to treat rheumatoid arthritis [78]. Wang et al. identified a derivative of sinomenine (YL064) as a potent inhibitor of STAT3 phosphorylation with anti-MM effects [79]. YL064 was shown to target the SH2 domain of STAT3 resulting in selective inhibition of Tyr705 but not Ser727 phosphorylation [79]. Importantly, YL064 continued to exert its pro-apoptotic effect when MM cells were co-cultured with stromal cells [79]. This data adds to the evidence supporting STAT3 inhibition as a means to overcome pro-survival signalling from the microenvironment.

Nifuroxazide was identified as a novel STAT3 inhibitor in a high throughput screen of candidate compounds [80]. This study showed nifuroxazide to be a selective inhibitor of STAT3 tyrosine phosphorylation which also inhibited the upstream kinases JAK2 and non receptor tyrosine protein kinase (TYK2). It was also shown that treatment with nifuroxazide leads to downregulation of MCL-1 [80]. It is not certain however whether the STAT3 inhibition reported here is a direct effect of nifuroxazide on STAT3 or a result of JAK2 inhibition.

Synthetic inhibitors targeting specific domains of STAT3 having been evaluated extensively against a variety of malignancies [81]. Among the synthetic STAT3 inhibitors, compounds targeting the SH2 domain have shown the most promise against MM [81]. LLL12, OPB-31121 and OPB-51602 showed potent in vitro activity against MM cells [82,83,84]. Unfortunately, they also showed significant toxicity and unfavourable pharmacokinetic profiles when evaluated in clinical trials which has hampered their development [81,83].

A number of other plant derived compounds have been reported to induce apoptosis in MM cells via inhibition of STAT3 [85,86,87,88,89,90,91,92,93,94,95]. Inhibition of upstream kinases JAK1,2, SRC and upregulation of SHP1 or other protein tyrosine phosphatases appear to be a common feature of these agents. Other protein tyrosine phosphatases activated include PIAS3 (induced by hydrocalamenene) phosphatase and tensin homolog (induced by Icariside II) and protein tyrosine phosphatase epsilon (induced by Arctiin) [89,92,93]. Among these agents, hydrocalaemenene, genipin, icariside II, arctiin and ursolic acid were shown to synergise in vitro with bortezomib and thalidomide while decursin synergised with doxorubicin [89,90,91,92,95]. The plant derived triterpenoid asiaticoside was shown to exert anti myeloma effects through induction of autophagy and activation of caspases in addition to reducing STAT3 phosphorylation [96]. Gossypol, a polyphenol derived from the cotton plant has also been shown to inhibit STAT3 phosphorylation via inhibition of IL-6 signalling [97]. Interestingly this compound also acts as a BCL2 homology domain 3 (BH3) mimetic, inducing apoptosis in MM cell lines, suggesting it has multiple mechanisms of anti MM activity. It is noteworthy that a number of these agents have been used historically in traditional medication, providing some evidence of their safety [88,89,91]. More systematic evaluation of the efficacy and toxicity profiles of these compounds in clinical trials is however called for. Given the promising in vitro data, their combination with proteasome inhibitor and IMIDs would be of interest.

## 5. Conclusions and Future Directions

Major advances have been made in our understanding of the role of STAT3 as well as JAK/STAT signalling in MM [10,81]. Although a variety of novel STAT3 inhibitors have been described, their lack of progress in clinical trials is disappointing. The chances of success with STAT3 inhibition in clinical trials maybe enhanced by better selection of patients who may benefit from STAT3 inhibition.

Given the rarity of STAT3 mutations in MM, developing a standardised biomarker to select patients for STAT3 inhibition maybe a challenge. Gene expression profiling to identify a STAT3 overexpression signature as described by Huang et al. in lymphoma maybe a useful strategy to identify patients for clinical trials [98].

The unfavourable pharmacokinetic profiles of some STAT3 inhibitors evaluated thus far has accounted for their failure in clinical trials [81]. The use of CD38 targeted nanoparticles loaded with STAT3 inhibitors shows promise as an innovative method to overcome this hurdle [99]. The development of more novel drug delivery techniques will be an important means of making STAT3 inhibition more clinically applicable.

Monoclonal antibodies against PRL-3 (PRL3-zumab) are currently in Phase I clinical trials [100], and targeting PRL-3 could be an attractive strategy to target STAT3 in MM. Synergism between PRL-3-zumab and PIs or immunomodulatory agents (IMIDs) also warrants further investigation.

Finally, the role of STAT3 in mediating microenvironment-derived survival signalling and treatment resistance provides an exciting opportunity as a therapeutic target. The acquisition of new genetic abnormalities and clonal evolution is key to the development of treatment resistance and may explain why MM remains incurable [101]. The prevalence of STAT3 overexpression in relation to the clonal architecture at serial time points during the patient’s disease would therefore be of great interest. As treatment resistance is often associated with loss of TP53, the question of whether STAT3 inhibition retains its potency in this setting would be particularly important to answer [102]. It is also interesting to speculate whether STAT3 inhibitors may augment the activity of current MM therapies in vivo. The role of STAT3 as a therapeutic target in MM is likely to evolve rapidly over the next few years. With more precise patient selection and optimisation of drug combinations, STAT3 inhibitors are likely to become a valuable addition to the expanding arsenal of drugs against MM.

## Figures and Tables

**Figure 1 cancers-11-00731-f001:**
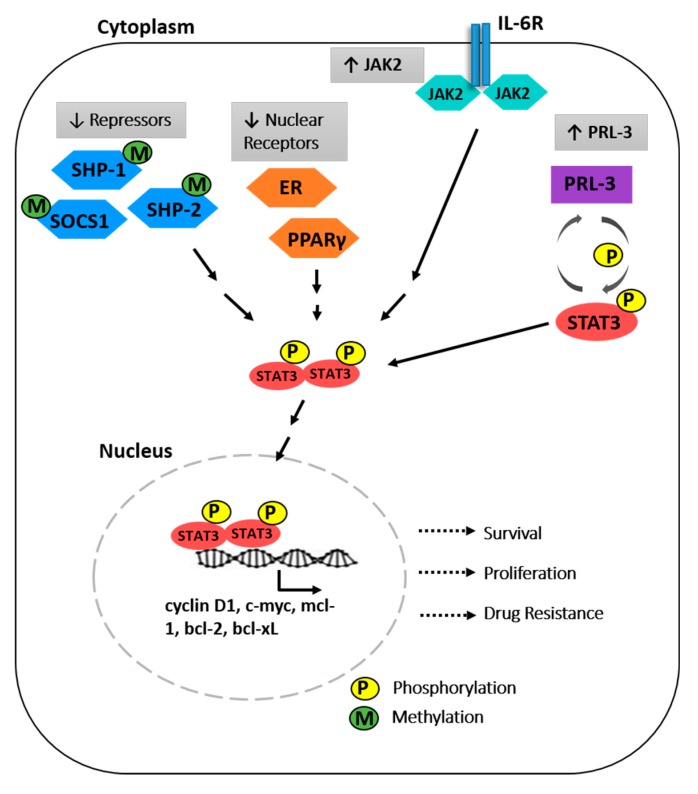
The molecular mechanisms driving constitutive signal transducers and activators of transcription (STAT3) activation in myeloma cells. Phosphorylated STAT3 translocate to the nucleus to mediate transcription of target genes, resulting in increased survival, proliferation and drug resistance of myeloma cells. Src homology containing protein 1 (*SHP-1*), suppressor of cytokine signalling 1 (*SOCS1*), phosphatase of regenerating liver 3 (PRL3), Janus Associated kinase (JAK), peroxisome proliferator activated receptors (PPAR), estrogen receptor (ER).

**Table 1 cancers-11-00731-t001:** Selected STAT3 inhibitors showing in vitro and/or clinical efficacy against multiple myeloma. The mechanism by which STAT3 is inhibited by the compound as well as evidence for in vitro synergy with known anti myeloma therapy and clinical evidence where applicable are presented. STAT3 (Signal transducers and activators of transcription 3), PIAS3 (protein inhibitor of activated STAT3), SHP-1 (SRC homology 2 domain containing phosphatase 1) PTEN (phosphatase and tensin homolog). MM (multiple myeloma), NA (data not available), SH2 (src homology 2), JAK (janus associated kinase), IL-6 (Interleukin 6).

STAT-3 Inhibitor	Mechanism of STAT3 Inhibition	In Vitro Synergy with Known Anti MM Agents	Clinical Evidence of Efficacy	Reference
Ruxolitinib	Indirect, via JAK inhibition	Bortezomib Lenalidomide	Phase 1 clinical trial	Chen et al. 2014 [70] Berenson et al. 2018 [71]
Tofacitinib	Indirect, via JAK inhibition	Venetoclax	NA	Lam et al. 2018 [72]
INCB16562	Indirect, via JAK1 inhibition	Bortezomib Melphalan	NA	Li et al. 2010 [74]
YL064	Direct, STAT3 SH2 domain inhibitor	NA	NA	Wang et al. 2018 [78]
OPB51602	Direct, STAT3 SH2 domain inhibitor	NA	Phase 1 clinical trial. Excessive toxicity and unfavourable pharmacokinetic profile	Ogura et al. 2015 [83]
Hydrocalamenene	Indirect, JAK1,2, SRC inhibition. Upregulation of PIAS3	Bortezomib	NA	Nam et al. 2014 [89]
Genipin	Indirect, SRC inhibition, SHP-1 upregulation	Bortezomib, thalidomide, paclitaxel	NA	Lee et al. 2011 [91]
Icariside II	Indirect, JAK2, SRC inhibition. Upregulation of SHP-1 and PTEN	Bortezomib, Thalidomide	NA	Kim et al. 2011 [92]
Niclosamide	Indirect inhibition by inhibiting IL-6 mediated phosphorylation of STAT3	NA	NA	Khanim et al. [77]
Asiaticoside	Reduced phosphorylation of STAT3, mechanism not known	NA	NA	Yingchun et al. [96]
Gossypol	Inhibition of IL-6 signalling	NA	NA	Sadahira et al. [97]
LCL161	Not known, synergism with JAK2 inhibitor against MM cell lines	NA	NA	Ramakrishnan et al. [76]

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
