# Peer review of "STAT3: A Promising Therapeutic Target in Multiple Myeloma"

_cancers, 2019, doi:10.3390/cancers11050731_

Round 1
Reviewer 1 Report
The review by Chong et al on STAT3 as a promising therapeutic target in MM is well articulated, detailed and thorough. This reviewer has minor comments and suggestions to improve the review.
1) It is well established that STAT3 translocates to mitochondria and alters cell functions esp. ETC and mitochondria membrane potential. The authors should include a paragraph on STAT3 effects in mitochondria in relation to MM.
2) The review introduction would also benefit by including post-translational modifications of STAT3 like SUMOylation, acetylation, S-nitrosylation, methylation, ubiquitination etc and other posttranslational modifications besides just Ser and Tyr phosphorylation.
3) The table1 indicating STAT3 inhibition strategies is non-exhaustive. There are several inhibitors used in MM in vitro and in vivo (albeit with failure) that should also be mentioned eg. niclosamide, Gossypol, LCL161, Asiaticoside etc.
4) STAT3 also has direct metabolic connection and regulates miRNA and long non-coding RNA in that regard. The authors should discuss the role of STAT3 in the context of MM.
5) The authors should comment on drug-resistant MM cells and if STAT3 targeting would be of benefit in that context?
6) The authors should comment on genetically heterogeneous MM and is STAT3 almost always activated and therefore a promising target?
Author Response
Reviewer #1
1) It is well established that STAT3 translocates to mitochondria and alters cell functions esp. ETC and mitochondria membrane potential. The authors should include a paragraph on STAT3 effects in mitochondria in relation to MM. (see section 2 paragraph 6)
We have included the following paragraph into the revised manuscript.
While the canonical functions of STAT3 as a transcriptional regulator are well-studied, findings that STAT3 can localize to the mitochondria (mitoStat3) in multiple cell types, including MM, revealed its regulatory activities in cellular metabolism. The interacting partners of STAT3 in the mitochondria are implicated in mitochondrial translation and electron transport chain (ETC) pathways. It was also reported that mitoSTAT3 is essential for Ras-mediated cellular transformation, and blockade of mitoSTAT3 leads to anti-tumor growth. Moreover, these processes were dependent on the phosphorylation of STAT3 on serine 727, which can be induced by IL-6. Given the importance of IL6-STAT3 signaling pathway in the survival of myeloma cells, it will be interesting to study mitoSTAT3 in the context of myeloma cells. Additionally, the role of STAT3 in mediating drug resistance was also suggested to be attributed to mitoSTAT3 through resistance to oxidative stress. Collectively, these findings suggest that targeting mitoSTAT3 should also be considered in addition to targeting the transcriptional activities of STAT3. In other studies, high levels of glutamine or miR551b in invasive ovarian cancer cells was found to activate STAT3, further implicating STAT3 in metabolism.
2) The review introduction would also benefit by including post-translational modifications of STAT3 like SUMOylation, acetylation, S-nitrosylation, methylation, ubiquitination etc and other posttranslational modifications besides just Ser and Tyr phosphorylation. (See section 2 paragraph 5)
We have included the following paragraph into the revised manuscript:
Apart from phosphorylation and methylation, other post-translational modifications like acetylation, SUMOylation, S-nitrosylation and ubiquitination are reported to regulate the functions of STAT3 in multiple cancer types. Acetylation of STAT3 has been reported to promote its transcriptional activity, especially on lysine 685. STAT3 also reportedly bind to PIAS proteins (SUMO E3 ligases), and the interaction can be promoted by cytokines like IL-6, implicating a role of SUMOylation in cytokine signaling. In MM cell lines, S-nitrosylation of STAT3 negatively regulates its activities resulting in cell cycle arrest and could rescue the sensitivity of cells towards melphalan. Activation of E3 ubiquitin ligase c-Cbl leads to the proteasomal degradation of SOCS3, promoting STAT3 activity. The relevance of these modifications in MM will therefore be of research interest.
3) The table1 indicating STAT3 inhibition strategies is non-exhaustive. There are several inhibitors used in MM in vitro and in vivo (albeit with failure) that should also be mentioned eg. niclosamide, Gossypol, LCL161, Asiaticoside etc.
Thank you for the comment. We have amended the table (see table 1 v3) to include some of the other STAT3 inhibitors suggested by the reviewer. Our aim with the table was to highlight to the reader, the agents we feel have the strongest evidence base and are most likely to progress to clinical trials, therefore we have not listed all the STAT3 inhibitors with anti myeloma activity reported in the literature.
4) STAT3 also has direct metabolic connection and regulates miRNA and long non-coding RNA in that regard. The authors should discuss the role of STAT3 in the context of MM.
Thank you for the comment, we have addressed this point in section 2 paragraph 6.
5) The authors should comment on drug-resistant MM cells and if STAT3 targeting would be of benefit in that context?
Thank you for the comment, we have addressed this in section 5 paragraph 5
6) The authors should comment on genetically heterogeneous MM and is STAT3 almost always activated and therefore a promising target?
Thank you for the comment, we have addressed this in section 2. Paragraph 1 and Section 5 paragraph 5.

Reviewer 2 Report
- page 1, rule 35 adverse cytogenetics: indeed amplification 1q is an adverse prognostic marker, however in omst studies del 17p t4;14 en t14;16 are used in analyses for high-risk cytogenetics. Maybe it is better to use these 3 as adverse risk examples.
Author Response
Thank you for the comment, we have addressed this in Section 1.1 Paragraph 1
Reviewer 3 Report
The work highlight the role of STAT3 and the JAK2 pathway and the role of PRL3 in the etiology and outcome of multiple myeloma an sheds the light on potential therapeutic interventions that are currently under trial or may be used as potential adjuvent therapeutic tools for MM.The authors discussed the optimization and the pharmacokinetic profiles of novel STAT3 inhibitors and highlighted that identifying how best to combine these agents with existing anti-myeloma therapy are key questions to be addressed in future clinical trials.
Author Response
Thank you for the comment